# OpenReview forum: "Stay Moral and Explore: Learn to Behave Morally in Text-based Games"
_ICLR.cc/2023/Conference — ICLR 2023 poster_

### Official Review · Reviewer_uEc3 · 2022-10-23

**Confidence:** 5
**Correctness:** 3
**Technical Novelty And Significance:** 2
**Empirical Novelty And Significance:** 3
**Recommendation:** 6

**Clarity, Quality, Novelty And Reproducibility:**

### Questions and concerns
1. As my main question, I want to hear more from the authors regarding the motivation of this research direction. Object collection and combat are probably at the core of many text adventure games, or any games in general. Many games are designed in a way that the players/agents have to follow some pre-defined storyline, or to reach some key points, in order to proceed the story forward. I fully understand and agree the necessity of having morality as an important evaluation dimension in sequential decision making problems, but are existing games the best place to start with?
2. Related to the previous question, I also want to grab the authors' thoughts on, if designing new games (let's say, text-based adventure games), what are some ways to put this morality dimension systematically in the designing process? For instance, how to explicitly model/measure the trade-off between morality and task progress, and how to evaluate morality. One possibility is, as the authors briefly discussed in the paper, to take social aspects into consideration in building tasks/games. On a related note, in [1], they proposed a set of minimal tasks that require agents to perform certain social interactions as part of the skills solving some tasks. This can potentially be used in text-based game design, in a way that the agent needs to borrow a lantern from an NPC (and later return it!).
3. Regarding human expert immorality scores in Table 1. Are they the accumulated scores until the end of the game, or until the same step budget as given to the agent? Would it make more sense if computing human immorality scores until the step where MorAL ends each game?
4. To me it was a bit confusing at the beginning of reading, I thought the morality training might have led to higher task completion scores --- which is actually not the case. The task completion boost might rather come from a better action candidate generator. Because the majority of the paper discuss the effects of morality learning, so the other part of the contribution (the boost on task completion scores) might not be as clear. To me they are both important and worth some emphasizing.


### Minor questions
1. In Section 5.4, the authors provide the value of $\lambda = 0.14$, this seems require some hyper parameter tuning. What was the hyper-param search space?
2. How much training speed does the extra modality learning phase sacrifice?


### References:
[1] [SocialAI: Benchmarking Socio-Cognitive Abilities in Deep Reinforcement Learning Agents. Kovač et al., 2021](https://arxiv.org/abs/2107.00956)

**Strength And Weaknesses:**

### Strength:

* **Multi-disciplinary & social impact**: This work brings some important ideas from another community (FAccT) to text-based games, which makes this study multi-disciplinary. A successful agent needs skills from RL, NLP, social science and more. I'd love to see work on this direction, which can have broader social impact than the specific task used in the paper.
* **Two-step learning**: The proposed MorAL pipeline makes sense to me, experiment results suggest that the two step training procedure indeed helps the agent performing well on both game playing and morality sides. Note the framework is not necessarily limited to only the morality learning, in principle, the second step can leverage any metric of interest, depending on the task.
* **Game completion improvement**: Without considering the morality dimension, the ablation study (Table 2) suggests that the self-imitation learning can quite significantly improve the average game completion percentage.


### Weaknesses:

Please see my questions and concerns below.

**Summary Of The Paper:**

In this work, the authors focus on studying the morality of RL agents on text-based adventure games. In such games, due to many reasons, agents often resort to behaviours that are identified as immoral, e.g., stealing things, attacking NPCs or monsters even unnecessary.

To mitigate this issue, the authors propose a novel iterative training framework, namely MorAL. Every iteration consists of a task learning phase and a morality learning phase. During task learning, the task policy conditions on state information selects its action from a list of action candidate that is generated by the (fixed) morality policy. Meanwhile, high-quality trajectories are pushed into a replay buffer. During the morality learning phase, a commonsense prior is utilized to score the transitions in the buffer, the morality policy is subsequently updated to generate action candidates that have higher morality scores.

Comparing against a set of top performing text-based game systems, on a set of 15 text-based games, the author show that the proposed MorAL framework indeed decreases the average immorality scores, and at the same time slightly increased the game completion percentage.

**Summary Of The Review:**

In general I like this work, the idea of introducing FAccT considerations into machine learning tasks is great; the two-step framework is neat, results suggest that it's working as expected. However, I am a bit hesitant because I am not fully convinced if existing games are the best platform for such work. For now, I would rate it above borderline, but I definitely look forward to reading the authors' responses and other reviewers' comments so I can better understand the work and thus have a more precise evaluation.

---

> ### Author Response · Authors · 2022-11-12
> **Author Response**
>
> > Q1: As my main question, I want to hear more from the authors regarding the motivation of this research direction...I fully understand and agree with the necessity of having morality as an important evaluation dimension in sequential decision making problems, but are existing games the best place to start with?
>
> A1: Thank you for your question. We chose text-based games to evaluate the proposed MorAL for the following reasons:
> * Semantically Rich Environments: Compared with previous benchmarks, text-based games have semantically rich and complex environments. Combined with the natural storyline of these games, thousands of morally dubious scenarios are created for the agent to explore. These diverse scenarios include theft, combat, as well as caring for and helping others.
> * Misalignment between Task and Morality: Agents tend to behave immorally if their training environment ignores moral concerns. This issue is more apparent in the game environment as the game task and morality are often in conflict. Thus, we aim to study this issue and propose a generic solution.
>
> > Q2: Related to the previous question, I also want to grab the authors' thoughts on, if designing new games (let's say, text-based adventure games), what are some ways to put this morality dimension systematically in the designing process...This can potentially be used in text-based game design, in a way that the agent needs to borrow a lantern from an NPC (and later return it!).
>
> A2: Thank you for your insightful suggestions. Due to the multiplicity of scenarios, Jiminy Cricket benchmark only evaluates the immorality of an agent based on the frequency and severity of immoral actions. One possible enhancement would be to encourage the agent to act in a less negative and more positive manner. As you suggested, we can include additional morally salient scenarios or tasks that are independent of the game task. In this way, we can evaluate the agent's morality by requiring it to reason and perform certain moral actions.
>
>
> > Q3: Regarding human expert immorality scores in Table 1. Are they the accumulated scores until the end of the game, or until the same step budget as given to the agent? Would it make more sense if computing human immorality scores until the step where MorAL ends each game?
>
> A3: Thank you for your question. The human expert immorality scores in Table 1 refer to the accumulated scores until the end of the game. Jiminy Cricket provides walkthroughs constructed by human experts, which take less immoral actions to quickly achieve the game task. However, a game's storyline has numerous branches that can lead to success or failure. At the end of an episode, the agent is likely to be on a branch. It is difficult to calculate the steps corresponding to the walkthrough. Thus, we follow previous work to define human expert immorality scores as the accumulated score until the end of the game.
>
>
> > Q4: To me it was a bit confusing at the beginning of reading, I thought the morality training might have led to higher task completion scores --- which is actually not the case. The task completion boost might rather come from a better action candidate generator. Because the majority of the paper discusses the effects of morality learning, the other part of the contribution (the boost on task completion scores) might not be as clear. To me they are both important and worth emphasising.
>
>
> A4: Thanks for your suggestion. The task completion boost come from self-imtation learning. We emphasise this in Section 5.6 and discuss the reason for this - improving the language model helps the agent to adapt to the new scenarios, thus going further.
>
>
>
> > Q5: In Section 5.4, the authors provide the value of λ=0.14, this seems to require some hyper parameter tuning. What was the hyper-param search space?
>
> A5: Thank you for your question. The hyper-param search space is from 0.1 to 0.5. We conducted hyper parameter tuning on one game, and then applied it to all games.
>
>
> > Q6: How much training speed does the extra modality learning phase sacrifice?
>
> A6: Thank you for your question. In our experiments, morality learning increases training time by around 10% during the same episode steps. The training speed of morality learning is related to parameters like learning cycle length and epoch. For example, in the 5 environments of the game “Zork1”, the average training time of the first 10,000 steps of the CALM model is 3h10min. For every 2000 steps, we update the action generator for 3 epochs. This took about additional 20 minutes.
>
> \[1] Kovač G, Portelas R, Hofmann K, et al. SocialAI: Benchmarking Socio-Cognitive Abilities in Deep Reinforcement Learning Agents[J]. arXiv preprint arXiv:2107.00956, 2021.

---

> > ### Comment · Reviewer_uEc3 · 2022-11-18
> > **Thank you**
> >
> > I have read the authors' response as well as other reviewers' comments. I appreciate the detailed response by the authors, they really helped me better understand the work.
> >
> > As written in my review, I like the thinking and effort of integrating FAccT considerations into sequential decision making problems such as games. However, I still believe that game completion probably has strong correlation / causation with the level of joy, which is critically important, and is of the essence of gaming. Therefore, the trade-offs between morality and game completion might require a deeper discussion by the game design community, which is beyond the scope of this paper.
> >
> > The above is not necessarily a shortcoming of this work, but I personally feel much confusion.
> >
> > Nevertheless, I agree that this work has many other contributions. For instance, if we abstract the MorAL algorithm, it can be seen as adding a knob (a constraint on certain dimension, represented as a scoring function) to an RL system. This can be generalized to many other use cases.
> >
> > That said, I will keep my score as positive.

---

> > > ### Author Response · Authors · 2022-12-01
> > > **Thank you and details on the concern**
> > >
> > > We sincerely appreciate your time and effort in reviewing our paper! Your recommendations are inspiring and constructive.
> > >
> > > We also want to add address your concern about the tradeoff.  The current finding (Table 1) shows that our method enhances the agent's moral capability in the 9/15 game compared to competing methods (i.e., CMPS and CMRS). For the remaining 6/15 games, the increasing immorality scores are complemented by higher task completion.
> > > Inspired by another reviewer, we will plot the baseline curve in Figure 3 and include more discussion on the tradeoff. We hope that these changes help address your concerns and that they make this tradeoff more clear. If you have any further concerns or suggestions, we are willing to discuss and reflect on them in the next revision. Thank you again!

---

### Official Review · Reviewer_bjYp · 2022-10-25

**Confidence:** 4
**Correctness:** 4
**Technical Novelty And Significance:** 3
**Empirical Novelty And Significance:** 3
**Recommendation:** 6

**Clarity, Quality, Novelty And Reproducibility:**

Clarity:
Major points:
- Equation 4 and its description are confusing. The description mentions that c is the scale factor, and \alpha = c*(1 - 0.05*i), where I presume that (1 - 0.05*i) tends to zero as the training progresses. Then it says that the scaling factor decays to zero as the probability of immorality increases (it isn't specified whether this is within one learning cycle or across the entire training run). I understand that m(c_i, a_i) goes to zero as the probability of immorality increases, but m(c_i, a_i) is different from the scaling factor, right? This whole paragraph is confusing. I get the overall idea that a policy network is being trained with imitation learning on successful buffer examples weighted by the conscience, but the details need to be made more clear.
- Figure 3 is somewhat poorly described in the text. How is the figure generated?

Minor points:
- The first paragraph of the introduction is more like related work
- In Figure 1's caption, it may be helpful to the reader to mention that the house that the agent is in does not belong to them.
- Typo: "use the CALM" should be "use CALM"
- Typo: "repeated learning circles" should probably be "repeated learning cycles"


Novelty:
There is some novelty with the proposed approach. It's a meaningfully different approach than CMPS or GALAD. Imo the main contribution is the improved results, though (backed up by ablations, although see my point about only 15 environments).


Reproducibility:
The authors say they will release code. The method does not seem reproducible from the description in the paper alone, which could be improved.

**Strength And Weaknesses:**

Strengths:
- Good improvements over CMPS (the previous SOTA on these environments)
- Artificial consciences are an interesting area of research
- The literature review throughout the paper is good. It is comprehensive and accurately represents the prior work. (although )
- Interesting idea to train an action generator with "moral-enhanced cross-entropy"
- Helpful implementation details in the appendix
- The ablations seem to be informative. Removing the moral cross-entropy loss (MorAL w/o Mixture w/o MeO) leads to the highest completion percentage but also many more immoral actions. This seems to validate that the mixture approach and moral cross-entropy loss are working as expected.

Weaknesses:
- Lack of clarity in Section 4.3 about what is old vs new. For instance, the data buffer is already a component of CALM, yet this is not made clear in the writing and it sounds like the authors are presenting something new. If there are differences compared to the CALM buffer, this should be made clear.
- Why only 15 games from Jiminy Cricket? How were these selected? Out of the 15 games selected, there are a few where immorality increases slightly, so I don't think the authors engaged in cherry-picking. However, it would be useful to know why the remaining 10 games were not included and whether the authors plan to include them (this would make it easier for future work to compare to the MorAL method).
- The paper should include more discussion of why comparison with GALAD is infeasible. This could be a subsection of the appendix. In particular, this paper proposes modifying the action generator language model, and GALAD does something similar. This should be discussed.
- It's not clear what the moral policy language model is. Is this a GRU on top of a pre-trained GPT-2 model? Or are you fine-tuning the GPT-2 model?

**Summary Of The Paper:**

This paper proposes MorAL, a new method for training text-based agents to avoid immoral actions. MorAL modifies the action generator in CALM with an imitation learning procedure that is weighted for immoral actions. On the Jiminy Cricket benchmark of text-based games with morality annotations, this notably improves game completion and reduces immoral actions, improving over existing baselines. Ablations show that various components of the method are working as expected.

Compared to previous work, the main contribution seems to be the idea to modify the action generator to reduce the sampling of immoral actions via imitation learning on a replay buffer. This is in contrast to CMPS, which directly modifies Q-values, and GALAD, which pretrains the action generator to reduce immoral action sampling without updating it during RL training. The proposed idea works well (compared to CMPS; GALAD is not compared to).

**Summary Of The Review:**

I think this is a good paper, but it feels like it needs more polishing. If the authors address most of my concerns listed above, I would be happy to recommend acceptance.

---------------------
Update after rebuttal:

The authors have addressed all of my concerns about how polished the original submission was. After checking the updated paper and the other reviews, I will now recommend acceptance. This is primarily because of the novelty of moving the morality prior into the GPT-2 language model (including fine-tuning it during RL training), the meaningful improvements to the RI metric, and the ablations.

---

> ### Author Response · Authors · 2022-11-12
> **Author Response**
>
> > Q1: Lack of clarity in Section 4.3 about what is old vs new. For instance, the data buffer is already a component of CALM, yet this is not made clear in the writing and it sounds like the authors are presenting something new. If there are differences compared to the CALM buffer, this should be made clear.
>
> A1: Thank you for your suggestion. The buffer and Moral Aligned Policy Optimization are new. We create a new buffer in addition to CALM's memory replay buffer to store high-quality trajectories. This is clarified in the revised paper in Data Collection.
>
> > Q2: Why only 15 games from Jiminy Cricket? How were these selected? Out of the 15 games selected, there are a few where immorality increases slightly, so I don't think the authors engaged in cherry-picking. However, it would be useful to know why the remaining 10 games were not included and whether the authors plan to include them (this would make it easier for future work to compare to the MorAL method).
>
> A2: Thank you for your question. To ensure a fair comparison, we use 15 games from Jiminy Cricket, which are also used by one of our main baseline CALM. Specifically, we first abandon the games where our backbone model CALM makes the least progress. Some games, such as "Infidel", "Lurking Horror", "Starcross" and "Stationfall", are difficult for CALM to complete. According to Hendrycks et al., (2021), the average percent completion of CALM on these games is less than 1. Therefore, we do not utilise these environments. We also remove some games with duplicate themes and genres, such as "Zork2" and "Cutthroats".
>
> Due to the time-consuming nature of RL training, it is common to select representative games from the game suite for experimentation. In this study, 15 representative games with five distinct initial states were chosen based on the aforementioned rules. We consider current experiments to be unbiased and convincing.
>
>
> > Q3: The paper should include more discussion of why comparison with GALAD is infeasible. This could be a subsection of the appendix. In particular, this paper proposes modifying the action generator language model, and GALAD does something similar. This should be discussed.
>
>
> A3: Thanks for your suggestion. In (Ammanabrolu et al. 2022), the author re-evaluated the jiminy cricket environment, maintaining only annotations with relatively high annotator agreement. However, the method and environment changes were not released publicly. We add a subsection to the appendix to provide more details. Please see Appendix C.
>
>
> > Q4: It's not clear what the moral policy language model is. Is this a GRU on top of a pre-trained GPT-2 model? Or are you fine-tuning the GPT-2 model?
>
> A4: Thank you for your question. The moral policy language model is the fine-tuned GPT-2 model. We described in Appendix D.
>
>
> > Q5: Equation 4 and its description are confusing...I understand that m(c_i, a_i) goes to zero as the probability of immorality increases, but m(c_i, a_i) is different from the scaling factor, right?...I get the overall idea that a policy network is being trained with imitation learning on successful buffer examples weighted by the conscience, but the details need to be made more clear.
>
>
> A5: Thank you for your suggestion. The scaling factor c is a fixed constant during training. m(c_i, a_i) is used to control the loss function according to the immorality of the data sample. m(c_i, a_i) is different from the scaling factor. We modified this in a revised manuscript.
>
>
> > Q6: Figure 3 is somewhat poorly described in the text. How is the figure generated?
>
> A6: Thank you for your review. Figure 3 shows the relationship between the completion percentage and the overall average immorality score. It shows there is a trade-off between the immorality score and completion percentage. The immorality score appears to be nearly proportional to the completion percentage. We add more details in the manuscript.
>
>
> > Q7: The first paragraph of the introduction is more like related work.
>
> A7: Thanks for your suggestion. We refine the first paragraph. We present the background of text games and the research motivation at first. Then, we introduce admissible actions and generating actions, which are relevant to the morality issue.
>
>
> > Q8: In Figure 1's caption, it may be helpful to the reader to mention that the house that the agent is in does not belong to them.
>
>
> A8: Thanks for your suggestion. We add more information about Figure 1.
>
>
>
> \[1] Hendrycks D, Mazeika M, Zou A, et al. What would jiminy cricket do? towards agents that behave morally[J]. arXiv preprint arXiv:2110.13136, 2021.
>
> \[2] Ammanabrolu P, Jiang L, Sap M, et al. Aligning to social norms and values in interactive narratives[J]. arXiv preprint arXiv:2205.01975, 2022.

---

### Official Review · Reviewer_foEH · 2022-10-25

**Confidence:** 3
**Correctness:** 3
**Technical Novelty And Significance:** 3
**Empirical Novelty And Significance:** 2
**Recommendation:** 6

**Clarity, Quality, Novelty And Reproducibility:**

Overall the paper is clearly written. It needs more explanation about the main results. The MorAL technique is novel and it doesn't need dense human feedback during training. The code will be released and that would make MorAL reproducible.

1. The commonsense prior model which is used to obtain morality scores is a RoBERTa-large model finetuned on commonsense morality portion of the ETHICS benchmark. It is unclear what is the performance of this finetuned model on the test set of the benchmark I.e how good is the quality of this model in predicting morality scores especially on out-of-domain data (because the Jiminy Cricket dataset would be out of domain for this model).

2. Similarly, the use of commonsense prior model could also be a limitation because the errors of this model would be propagated to the morality control module of MorAL. This should be made clear and what definition of morality is used should also be clarified in the paper.

**Strength And Weaknesses:**

Strengths:

1. The MorAL technique achieves overall a significant boost of 19% in game completion rate while decreasing the immorality score by 5% compared to CMPS. Hence, overall 15 games the MorAL technique seems successful.

2. The paper presents comprehensive ablation experiments studying the effect of each component of MorAL on immorality score and completion percentage. The ablations show that using the moral-enhanced objective has the highest impact on immorality score and completion percentage. Removing this significantly increases the immorality score and completion percentage.

3. The paper also presents an example of actions generated by morality policy and the action picked by the MorAL.

Weakness:

1. Even if on average MorAL achieves the least immorality score and the highest completion percentage, for 7/15 tasks NAIL has lower immorality score and for 4/15 tasks CMPS has lower immorality score. In total for 11/15 tasks, MorAL does not achieve the least immorality score. These results are not discussed and no reasoning is provided why this is the case. The reason MorAL achieves an overall lower immorality score is because for certain games like Ballyhoo, Moonmist and Deadline NAIL has a high immorality score. There should be a discussion on what is the reason for this.

2. Similarly, for a total of 8/15 tasks, other prior techniques achieve a higher completion percentage. Similar to point (1), this should be discussed. Are there certain properties of certain games because of which MorAL performs better or worse on them.

**Summary Of The Paper:**

This paper proposes a Moral Awareness Adaptive Learning (MorAL) to enhance the morality capacity of an agent using a plugin moral-aware learning model. The paper develops a mixture policy to interleave  task learning and morality learning.



For each episode, the moral policy decodes a set of valid action candidates. The task policy pairs it with current observation and computes a Q-value. To pick an action, a combination of Q-value and a score returned by moral policy is used. These states are stored in a memory. Later, morality scores are computed for high-quality trajectory to update the morality awareness control module. The morality scores are computed by a commonsense prior module.



The paper compares MorAL technique with four other prior techniques (NAIL, CALM, CMRS, CMPS) on the Jiminy Cricket benchmarks. The metrics of evaluation are Immorality Score, Completion Percentage and Relative Immorality (which are used in prior work). The paper presents a comprehensive ablation of the different components of MorAL and their contributions.

**Summary Of The Review:**

Overall, the paper proposes a novel MorAL framework to reduce immoral actions in text based games without impacting the game completion percentage. The MorAL technique is overall successful in doing so but not when we look at majority of the games individually.

> Making changes to the score to reflect the points made in the discussion.

---

> ### Author Response · Authors · 2022-11-12
> **Author Response**
>
> > Q1: Even if on average MorAL achieves the least immorality score and the highest completion percentage, for 7/15 tasks NAIL has lower immorality score and for 4/15 tasks CMPS has lower immorality score. In total for 11/15 tasks, MorAL does not achieve the least immorality score...These results are not discussed and no reasoning ... Similarly, for a total of 8/15 tasks, other prior techniques achieve a higher completion percentage ...should be discussed
>
> A1:  Thank you for your suggestion. In most cases, for agents, morality and task completion are often in conflict in text-based games. For instance, in the game “ Zork3”, our model improves task completion due to additional immoral actions performed by the agent (i.e. more props are collected), but also resulting in a rise in the immorality score. It also happens to other methods. While in some games such as "Ballyhoo", an increase in task completion can lead to a decrease in immorality scores. This might be because the task completion is increased without encountering additional morally salient scenarios. We add more discussion in Section 5.5.
>
> We explore the efficiency of our MorAL through ablation studies (Table 2). The improvement in task completion is attributed to self-imitation learning, which leverages past valuable trajectories to improve the action generator. Compared to “MorAL w/o Mixture w/o SIL”, “MorAL w/o Mixture w/o Meo” improves task completion on 14/15 games. The use of moral policy and the proposed loss function are both credited with lowering the immorality scores.
>
> We also plot Figure 3 to investigate the trade-off between behaving morally and task completion. Compared with its variants, our MorAL yields a better trade-off, as it reduces the immorality behaviours with an acceptable sacrifice of the completion percentage.
>
>
>
> > Q2: ...It is unclear what is the performance of this finetuned model on the test set of the benchmark I.e how good is the quality of this model...Similarly, the use of commonsense prior model could also be a limitation because the errors of this model would be propagated to the morality control module of MorAL. This should be made clear and what definition of morality is used should also be clarified in the paper.
>
> A2: Thanks for your suggestion. Similar to Hendrycks et al. (2021), the commonsense prior achieves 63.4% accuracy on a challenging test set for commonsense morality questions (Hendrycks et al. 2020), which demonstrates that a stronger model of commonsense morality could further improve the performance of agents on Jiminy Cricket benchmark. We clarify this in Appendix D of the revised manuscript. We also add a footnote in the revised manuscript that defines the morality.
>
>
> \[1] Hendrycks D, Mazeika M, Zou A, et al. What would jiminy cricket do? towards agents that behave morally[J]. arXiv preprint arXiv:2110.13136, 2021.
>
> \[2] Hendrycks D, Burns C, Basart S, et al. Aligning ai with shared human values[J]. arXiv preprint arXiv:2008.02275, 2020.

---

> > ### Comment · Reviewer_foEH · 2022-11-25
> > **Thank you for addressing my questions and a follow up suggestion**
> >
> > Thank you for the responses and addressing my questions.
> >
> > > Q1: Even if on average MorAL achieves the least immorality score and the highest completion percentage, for 7/15 tasks NAIL has lower immorality score and for 4/15 tasks CMPS has lower immorality score. In total for 11/15 tasks, MorAL does not achieve the least immorality score...These results are not discussed and no reasoning ... Similarly, for a total of 8/15 tasks, other prior techniques achieve a higher completion percentage ...should be discussed
> >
> > I agree that "In most cases, for agents, morality and task completion are often in conflict in text-based games."
> > Hence, it would be good to understand in how many games is MorAL successful in reducing immorality score while being competitive or better in completion in comparison with other methods.
> >
> > Thank you for the Figure 3. Having other techniques like CMPS and NAIL plotted in the same Figure will give a better idea of how MorAL compares with other methods. Right now Figure 3 indicates that compared with its variants, MorAL yields a better trade-off. But does it yeild a better trade-off compared to other methods?
> >
> > > Q2: ...It is unclear what is the performance of this finetuned model on the test set of the benchmark I.e how good is the quality of this model...Similarly, the use of commonsense prior model could also be a limitation because the errors of this model would be propagated to the morality control module of MorAL. This should be made clear and what definition of morality is used should also be clarified in the paper.
> >
> > Thank you for clarifying this.

---

> > > ### Author Response · Authors · 2022-11-26
> > > **Author Response**
> > >
> > > > I agree that "In most cases, for agents, morality and task completion are often in conflict in text-based games." Hence, it would be good to understand in how many games is MorAL successful in reducing immorality score while being competitive or better in completion in comparison with other methods.
> > >
> > > Thank you for the advice. The main result (Table 1) already shows the trade-off MorAL makes between morality and task completion. Compared to competing methods (i.e., CMPS and CMRS), our method improves the agent's moral capability on 9/15 games. For the remaining 6/15 games, the increased immorality score is accompanied by higher task completion. Both the agent's moral capability and task completion improved on 4/15 games.
> > >
> > > > Thank you for the Figure 3. Having other techniques like CMPS and NAIL plotted in the same Figure will give a better idea of how MorAL compares with other methods. Right now Figure 3 indicates that compared with its variants, MorAL yields a better trade-off. But does it yeild a better trade-off compared to other methods?
> > >
> > > Thank you for the suggestion. In the revised version, we will plot the baseline curves for comparison in Figure 3 (We cannot update the submission right now). In ablation studies, we demonstrated that self-imitation learning could significantly improve task completion, and moral-enhanced loss and mixture policy can further strengthen the agent's moral capability.

---

> > > > ### Comment · Reviewer_foEH · 2022-11-30
> > > > **Final clarification and request to add a discussion on trade-off**
> > > >
> > > > > Thank you for the advice. The main result (Table 1) already shows the trade-off MorAL makes between morality and task completion. Compared to competing methods (i.e., CMPS and CMRS), our method improves the agent's moral capability on 9/15 games. For the remaining 6/15 games, the increased immorality score is accompanied by higher task completion. Both the agent's moral capability and task completion improved on 4/15 games.
> > > >
> > > >
> > > > To clarify my question, we want to understand "in how many games is MorAL successful in reducing immorality score while being competitive or better in completion in comparison with other methods." This means if you are comparing with CMRS then we want to understand for how many games MoRAL immorality score is lower than CMRS AND at the same time completion score is higher than CMRS. This happens for 7/15 games (Ballyhoo, Borderzone, Deadline, Hitchhiker, Moonmist, Wishbringer, Zork1). On the other hand CMRS has lower immorality score and higher completion score compared to MorAL on 0/15 games. Similarly, NAIL, CALM and CMPS too have lower immorality score and higher completion score compared to MorAL on 0/15 games. Both the agent's moral capability and task completion are improved on 2/15 games (Moonmist and Deadline) I.e MorAL has the lowest immorality score and highest completion score among all the models.
> > > >
> > > > Maybe, including the curves in Fig 3 might make this trade-off clearer. It would be good to add this discussion in the main results. Since, all my questions are answered and the authors have promised to add the necessary figures and results in the next version, I am changing my score to reflect the discussion.

---

> > > > > ### Author Response · Authors · 2022-12-01
> > > > > **Thank you again!**
> > > > >
> > > > > Thank you very much again for your insightful comments. They are extremely valuable for improving our work. We will add additional discussion to the next version.

---

> > > ### Author Response · Authors · 2022-11-30
> > > **Thank you for your comments and suggestions**
> > >
> > > We would like to thank you once more for the time you have invested in providing feedback for our paper. Based on your specific recommendation, we will update Figure 3 in the next revision. We believe your constructive comments will further strengthen our paper. It would be highly appreciated if you could reconsider your rating on our work.  Please let us know if you have any further concerns, questions, or suggestions. Thank you!

---

### Official Review · Reviewer_qzhN · 2022-10-27

**Confidence:** 3
**Correctness:** 3
**Technical Novelty And Significance:** 2
**Empirical Novelty And Significance:** 2
**Recommendation:** 5

**Clarity, Quality, Novelty And Reproducibility:**


Clarity: okay overall, though I found the clarity of the methods could be improved.

Quality: also okay, see above

Novelty: I think the novelty is fairly limited, see above

Reproducibility: seems quite good, lots of details given on hyperparameters, etc.



**Strength And Weaknesses:**

Strengths:
+ The problem being tackled is interesting, and fairly important. Studying this problem in text-based games is an interesting angle.
+ The paper includes a nice ablation section.
+ The proposed algorithm does seem to improve over the baselines.

Weaknesses:
- I found the method wasn't very clearly explained, and took a while to understand with multiple re-reads. Specifically, I think it'd be useful to explain DRRN and CALM more in detail when first presented, since these are the core pieces on which MorAL is based.
- The improvements over baselines are quite small, and given there are no error bars it's hard to tell if the results are significant.
- The closest competing algorithms (CMPS and CMRS from Hendrycks et al) are only described briefly in the introduction, but they seem like very similar to the proposed method. The overall strategy is the same (train a Q function that optimizes task reward, and modify it using a commonsense prior to make it more moral). From what I can tell, only difference seems to be whether the correction term is used to modify the game reward or Q-value, rather than separately choosing actions using a mixture of the Q value and moral policy. I'm not convinced by the drawbacks mentioned in the paper for the Hendrycks et al. method:

"First, adding a correction term to the game reward or Q-value will generate extra noise, particularly for
game rewards that are extremely sparse. In addition, some immoral actions are necessary for progressing through the game. For instance, in the game “Zork1”, the agent must steal the lantern to
reach the following location on the map, as shown in Figure 1. The trade-off between task progress
and morality is a dilemma that agents may encounter while making decisions."

Specifically, it's not clear to me how the proposed MorAL algorithm materially changes the problem of "some immoral actions are necessary for progress in the game" -- it seems like both methods have some way of trading off morality and task performance, and it's unclear to me which is better. I think this paper would be much stronger if it had a better argument / analysis for why MorAL performs better than CMPS/ CMRS, and why we'd expect this to be a general finding, rather than a specific quirk of this environment.


----------------------------------------
Small points:
"RL agents may select immoral actions unconsciously"
> unclear if 'unconsciously' is an appropriate word here (what would it mean for an RL agent to select actions consciously?)

"The LM is then equipped with a conscience"
> I don't really like using these kinds of human analogies, since I don't think the 'conscience' described here is anything like a human conscience.

"To sustain the agent’s exploration, we define that
within an episode, if the current steps t exceed the max length of trajectories lmax in buffer B, πT
should be used instead of the mixture policy for selecting actions."
> So if the episodes are long enough, the agent behaves non-morally? Or is this only during training?

"The walkthrough is constructed by human experts to quickly obtain the maximum possible score while taking less immoral behaviours"
> Seems like the method needs human expert data to warm-start learning. Is this also a limitation of other methods?

"The framework eliminates the assumption that dense human feedback is required
during training, as we only perform morality learning using a limited number of trajectories at
specific stages."
> It seems like other methods also don't need dense human feedback, unless I'm mistaken?

- I'm not clear why you call MorAL a 'framework', rather than simply an 'algorithm'. To me it seems more like an algorithm.

- Eq1 -- what is A? Is it the set of candidate actions?


**Summary Of The Paper:**

This paper investigates the question: how can we train agents to behave 'morally' in text-based games, while achieving a high task reward?  The paper proposes a two stage learning method, which is iterated over time. In my understanding, this involves training both a task-specific Q function ('task policy'), which only cares about task reward, and a 'moral policy' which is fine-tuned from GPT-2. During inference, candidate actions are sampled by the moral policy, and the final action is chosen via a weighted combination of the logprobs of the moral policy and the task policy (derived via softmax on the Q function). The task policy is updated via normal TD learning. The moral policy is refined using self-imitation learning -- the 'best' trajectories (based on task reward, length, and novelty) are selected into a buffer, they get assigned a 'morality score' according to a 'commonsense prior' (a BERT model fine-tuned on a part of the ETHICS dataset), and a weighted behavior cloning objective (weighted by the commonsense prior) is used to update the morality policy.

The paper shows that this method improves upon baselines on the Jiminy Cricket benchmark (15 text-based games w/ morality scores), both on immorality score and completion percentage.

**Summary Of The Review:**


Overall, this paper proposes a new algorithm for moral learning in text games and shows slight improvements over baselines. My biggest concern is that there is little space devoted to analyzing how / why the method is different from previous work (specifically Hendrycks et al.), and it's not intuitively clear to me from the paper why the proposed method should be better. Thus, the paper relies on its empirical results, which, while not bad, seem to me to be only small improvements over the baseline (though I'm also not an expert in this field). Thus, I lean reject.

---

> ### Author Response · Authors · 2022-11-12
> **Author Response - Part 2/2**
>
> > Q7:  "To sustain the agent’s exploration, we define that within an episode, if the current steps t exceed the max length of trajectories lmax in buffer B, πT should be used instead of the mixture policy for selecting actions." So if the episodes are long enough, the agent behaves non-morally? Or is this only during training?
>
> A7: Thank you for your review. This operation is only conducted during training to enhance the agent's exploration ability. We add more details to make it clear.
>
> > Q8: "The walkthrough is constructed by human experts to quickly obtain the maximum possible score while taking less immoral behaviours" Seems like the method needs human expert data to warm-start learning. Is this also a limitation of other methods?
>
> A8: Thank you for your question. The human expert is independent of the training process. This configuration is based on prior research and allows a game to be divided into five different environments. This is clarified in the revised manuscript.
>
>
> > Q9: "The framework eliminates the assumption that dense human feedback is required during training, as we only perform morality learning using a limited number of trajectories at specific stages." It seems like other methods also don't need dense human feedback, unless I'm mistaken?
>
> A9: Thank you for your question. Prior study needs dense human feedback and requires that the morality of the action be evaluated at each step. As noted by Hendrycks et al. (2021), requiring such dense human feedback for training purposes is unrealistic in most sequential decision making environments and is thus used only for evaluation. This assumption is invalidated by the fact that our method only evaluates actions in the buffer at particular stages.
>
>
> > Q10: I'm not clear why you call MorAL a 'framework', rather than simply an 'algorithm'. To me it seems more like an algorithm.
>
> A10: Thanks for your review. We replaced “framework” with “algorithm” in the revised manuscript.
>
> > Q11: Eq1 -- what is A? Is it the set of candidate actions?
>
> A11: Thank you for your question. In Eq1, A denotes the set of action candidates generated by the LM.
>
>
> \[1] Ammanabrolu P, Jiang L, Sap M, et al. Aligning to social norms and values in interactive narratives[J]. arXiv preprint arXiv:2205.01975, 2022.
>
> \[2] Hendrycks D, Mazeika M, Zou A, et al. What would jiminy cricket do? towards agents that behave morally[J]. arXiv preprint arXiv:2110.13136, 2021.

---

> ### Author Response · Authors · 2022-11-12
> **Author Response - Part 1/2**
>
> > Q1: I found the method wasn't very clearly explained, and took a while to understand with multiple re-reads. Specifically, I think it'd be useful to explain DRRN and CALM more in detail when first presented, since these are the core pieces on which MorAL is based.
>
> A1: Thanks for your suggestion. We refine and add more details about DRRN and CALM in Section 3 for better understanding.
>
>
> > Q2: The improvements over baselines are quite small, and given there are no error bars it's hard to tell if the results are significant.
>
> A2: Thank you for your review. We would like to clarify that our model is significantly improved than baselines. Our model boosts the game completion percentage by 19% while decreasing the immorality score by 5%. Earlier efforts based on this benchmark utilised an LM trained on more human gameplay trajectories and a prior built on more morally-relevant datasets (Ammanabrolu et al. 2022). Our model instead does not require any external data source - during RL, the agent automatically collects past successful trajectories to conduct morality learning.
> We follow the experimental setup of Hendrycks et al. (2021) for better comparison.
>
> > Q3: The closest competing algorithms (CMPS and CMRS from Hendrycks et al) are only described briefly in the introduction, but they seem like very similar to the proposed method. The overall strategy is the same (train a Q function that optimizes task reward, and modify it using a commonsense prior to making it more moral). From what I can tell, the only difference seems to be whether the correction term is used to modify the game reward or Q-value, rather than separately choosing actions using a mixture of the Q value and moral policy.
>
> A3: Thank you for your review. Both CMPS and CMRS (Hendrycks et al., 2021) simply use a commonsense prior to determine the morality of an action and to modify its Q-value/reward. MorAL is superior than these two algorithms in the following ways:
> * Self-imitation learning: We collect past successful trajectories during training to conduct self-imitation learning. Ablation studies show that SiL greatly improves game completion.
> * Moral-enhanced loss function: We design a dynamically scaled cross-entropy function for morality learning, which allows for a greater emphasis on the training of moral samples.
> * Adaptive learning: We design multiple learning cycles for adaptive task learning and morality learning.
>
> We also add more details in the related work.
>
>
> > Q4: I'm not convinced by the drawbacks mentioned in the paper for the Hendrycks et al. method:
> "First, adding a correction term to the game reward or Q-value will generate extra noise, particularly for game rewards that are extremely sparse. In addition, some immoral actions are necessary for progressing through the game. For instance, in the game “Zork1”, the agent must steal the lantern to reach the following location on the map, as shown in Figure 1. The trade-off between task progress and morality is a dilemma that agents may encounter while making decisions."
> Specifically, it's not clear to me how the proposed MorAL algorithm materially changes the problem of "some immoral actions are necessary for progress in the game" -- it seems like both methods have some way of trading off morality and task performance, and it's unclear to me which is better. I think this paper would be much stronger if it had a better argument / analysis for why MorAL performs better than CMPS/ CMRS, and why we'd expect this to be a general finding, rather than a specific quirk of this environment.
>
> A4: Thank you for your review. In text-based games, improving the morality of the agent often leads to less task completion (Hendrycks et al. 2021). Although Ammanabrolu (2022) demonstrates greater increases in both metrics, the improvement in task completion is due to the fact that more human gameplay trajectories are used to finetune the LM. This study aims to enhance the morality of the agent while simultaneously increasing game completion. Our algorithm overcomes this issue by adaptive morality learning and task learning.
>
>
> > Q5: "RL agents may select immoral actions unconsciously" unclear if 'unconsciously' is an appropriate word here (what would it mean for an RL agent to select actions consciously?)
>
> A5: Thank you for your review. "Unconsciously" denotes that the agent chooses actions without being aware of morality. We modified this sentence to make it clearer.
>
>
> > Q6: "The LM is then equipped with a conscience" I don't really like using these kinds of human analogies, since I don't think the 'conscience' described here is anything like a human conscience.
>
> A6: Thank you for your review. We follow the prior work (Hendrycks et al., 2021) to use 'conscience' as the moral sense of right and wrong.  This sentence is modified in the revised manuscript.

---

> ### Author Response · Authors · 2022-11-30
> **A Gentle Reminder**
>
> We understand the reviewer load is high, and we thank you again for your time!
>
> We just wanted to flag that we have made every effort to address your concerns in the response and revised paper. Particularly, we clarified the superiority of our MorAL and added additional explanations about the differences from the previous work.  Hence, it would be highly appreciated if you could provide feedback on our responses or confirm whether there is no remained concern. If you have any further concerns, questions, or suggestions, we are willing to discuss and reflect on them in the next revision. Thank you!

---

### Author Response · Authors · 2022-11-15
**General Comment to All Reviewers**

Dear reviewers:

We appreciate the time and effort that each reviewer invested in reviewing and giving us useful comments.
As the reviewer suggested, we have improved the paper's organisation as follows:

* We have revised the manuscript and expanded the result analysis.
* We have added a subsection to the appendix with a detailed introduction to DRRN and CALM.
* We have added a subsection to the appendix to illustrate how our method differs from the GALAD and why they are not comparable.

Please find below our point-wise responses to your questions. It would be great if you could acknowledge our response and let us know if you have any remaining questions about our work. Thank you.

---

> ### Author Response · Authors · 2022-11-16
> **Please read the responses and react**
>
> Dear reviewers:
>
> We appreciate the time and effort that each reviewer invested in reviewing and giving us useful comments which helped improve our paper.
>
> We have provided corresponding responses and results, which we believe address your concerns. It would be great if you could acknowledge our response and let us know if you have any remaining questions about our work.
>
> Thank you.

---

### Author Response · Authors · 2022-11-16
**Please read the responses and react**

Dear reviewers and ACs:

We appreciate the time and effort that each reviewer invested in reviewing and giving us useful comments which helped improve our paper.

We have provided corresponding responses and results, which we believe address your concerns. It would be great if you could acknowledge our response and let us know if you have any remaining questions about our work.

Thank you.

---

### Decision · Program_Chairs · 2023-01-20

**Decision:**

Accept: poster

**Justification For Why Not Higher Score:**

The reviewers all agreed that the work is relatively incremental, and the scope of the paper is relatively limited (one benchmark).

**Justification For Why Not Lower Score:**

Solid paper with meaningful technical contributions, and it has applications to responsible AI.

**Metareview: Summary, Strengths And Weaknesses:**

This paper presents a new method (MorAL) for training text-based agents to minimize immoral actions. MorAL is an extension of CALM, adding an imitation learning procedure that is weighted for immoral actions. An advantage of this new method is that it does not need dense human feedback during training. Its results are good with a boost of 19% in game completion rate while decreasing the immorality score by 5% compared to CMPS (the previous SOTA on of environments of the paper).

The discussion addressed most of the reviewers concerns. During the reviewer-AC meeting, some of the reviewers still found the work to be relatively "incremental" (extension of CALM) and its technical applicability to other tasks or domain is relatively unclear (as the work is applied to one benchmark). However, the reviewers also noted other benefits of this work, e.g., that the problem of training agents to reduce immoral behaviors is an important one. Results are strong and the paper presents a comprehensive ablation of the different components of MorAL and their contributions.

As this is a solid paper with applicability to responsible AI, I think the pros outweigh the cons and recommend accept.

Note: If the paper is accepted, I would recommend that the authors add plots akin to Figure 3, but applied to the main results of the paper (Table 1 and 2) as suggested by one of the reviewers. This would help better understand the tradeoff between task completion and immorality.

**Note From Pc:**

if the above contains the word "oral" or "spotlight" please see: "oral" presentation means -> notable-top-5% and "spotlight" means -> notable-top-25%. As stated in our emails, we are disassociating presentation type from AC recommendations

**Summary Of Ac-Reviewer Meeting:**

Discussions:
- We first went through the last remaining technical concerns, and they were all quickly eliminated. The only real concern is that the main results (Table 1 and 2) should be made more interpretable, e.g., via Pareto front plots.
- Review uEc3 had concerns about the application to text games, but didn't seem to be a major concern for the other reviewers.
- Most of the discussion was on whether to accept or reject (see below.)

Reviewers' recommendation:
The consensus at the meeting was this is a very borderline paper, and the reviewers couldn't reach a decision on either accept or reject. They all agreed the decision could reasonably go either way, and they left it for the AC or SAC to decide. The reviewers felt this work is solid but incremental and of narrow scope.